

# DHA content in milk and biohydrogenation pathway in rumen: a review

Guoxin Huang[1,2,*], Yangdong Zhang[1,*], Qingbiao Xu[3], Nan Zheng[1], Shengguo Zhao[1], Kaizhen Liu[1], Xueyin Qu[4], Jing Yu[4] and Jiaqi Wang[1]

[1] Chinese Academy of Agricultural Sciences, State Key Laboratory of Animal Nutrition, Institute of Animal Science, Beijing, China
[2] Northeast Agricultural University, College of Animal Sciences and Technology, Harbin, China
[3] Huazhong Agricultural University, College of Animal Sciences and Technology, Wuhan, China
[4] Tianjin Mengde Groups Co., Ltd, Tianjin, China
[*] These authors contributed equally to this work.

## ABSTRACT

Docosahexaenoic acid (DHA) is an essential human nutrient that may promote neural health and development. DHA occurs naturally in milk in concentrations that are influenced by many factors, including the dietary intake of the cow and the rumen microbiome. We reviewed the literature of milk DHA content and the biohydrogenation pathway in rumen of dairy cows aim to enhance the DHA content. DHA in milk is mainly derived from two sources: $\alpha$-linolenic acid (ALA) occurring in the liver and consumed as part of the diet, and overall dietary intake. Rumen biohydrogenation, the lymphatic system, and blood circulation influence the movement of dietary intake of DHA into the milk supply. Rumen biohydrogenation reduces DHA in ruminal environmental and limits DHA incorporation into milk. The *fat-1* gene may increase DHA uptake into the body but this lacks experimental confirmation. Additional studies are needed to define the mechanisms by which different dietary sources of DHA are associated with variations of DHA in milk, the pathway of DHA biohydrogenation in the rumen, and the function of the *fat-1* gene on DHA supply in dairy cows.

## INTRODUCTION

Some researchers reported that milk fat could increase triglycerides in blood. However, many researches showed the milk fat had no adverse effects on the concentrations of fasting blood lipids, glucose, and insulin (*Benatar, Sidhu & Stewart, 2013*; *Engel, Elhauge & Tholstrup, 2017*). Some studies even showed the benefit for milk fat as a blood pressure supplement (*Rietsema et al., 2019*). Substances found in enriched milk, including medium and odd chain SFA (saturated fatty acid), globular phospholipids, unsaturated fatty acids, branched-chain fatty acids, natural trans fatty acids, vitamins K1 and K2, and calcium,

Corresponding author
Jiaqi Wang, wangjiaqi@caas.cn

have been found to have positive health effects (*Mozaffarian & Wu, 2018*). Among them, conjugated linoleic acid (CLA), which is peculiarly originated from the rumen (*Jaglan et al., 2019*), and omega-3 polyunsaturated fatty acids (n-3 PUFA) have been found to show health benefits to humans (*Swanson, Block & Mousa, 2012*).

Docosahexaenoic acid (DHA, C22:6n-3) is an n-3 PUFA found in the mammalian central nervous system (*Gázquez, 2017*), making up 10% to 15% of the total cerebral fatty acids (about 10 $\mu$mol/g brain). Some bacteria and lower eukaryotes can produce DHA *de novo* via a polyketide synthase pathway (*Kabeya et al., 2018*) but humans lack the key fat desaturase enzyme for synthesizing DHA (especially $\Delta 12$ and $\Delta 13$/n-3 desaturase). The *FAO/WHO (2008)* recommended a daily intake of DHA+EPA of 300 mg for lactating women, recent studies shown a daily intake of 100 mg DHA for infants and 250 mg/day for adolescents DHA+ Eicosapentaenoic acid (EPA, C20:5n-3) (*Saini & Keum, 2018*). The American Heart Association recommends an intake of 2–4 g/day of DHA+EPA for hypertriglyceridemia patients (*Miller et al., 2011*). *Gebauer et al. (2006)* recommended an intake of approximately 500 mg/d of EPA+DHA to reduce the risk of cardiovascular disease. However, most populations only get approximately 100 mg of DHA+EPA per day, which is much lower than the recommendations (*Afshin et al., 2019*).

The human body can synthesize DHA in extremely limited amounts using $\alpha$-linolenic acid (ALA, C18:3n-3) (*Plourde & Cunnane, 2007*), and only approximately 0–4% of dietary ALA may be converted to DHA (*Burdge & Wootton, 2002*), so DHA needs to be supplemented (*Hashimoto et al., 2017*).

Milk is a possible dietary source of DHA, dietary source of DHA, but its concentration is particularly low (*Bai et al., 2018*; *Ishaq & Nawaz, 2018*; *Shingfield, Bonnet & Scollan, 2013*). The DHA content of milk is influenced by the rumen microbiota, endogenous synthesis, and dietary intakes of DHA by dairy cows. The rumen biohydrogen content limits the efficient dietary incorporation of DHA into milk and the pathway of DHA hydrogen in the rumen is still unclear. Fat-1 gene was also used to increase DHA in milk (*Wu et al., 2012*), but researches is limited. We mainly analyzed the literature and defined factors affecting the conversion of dietary DHA into milk and explored strategies to increase the DHA content in milk.

## METHODOLOGY

The scholarly articles in this review were obtained from web of knowledge, google scholar Baidu scholar and subject-specific professional websites, the date from 1999-2019. The keywords "dairy cow", "dairy cattle", "rumen", "bacteria", "biohydrogenation" "DHA", "microalgae" and "fish oil" were used in the search. All of the articles included in this review were peer-reviewed. The article chose in this paper should show the relation between DHA with dairy cow bacteria or rumen biohydrogenation. The qualitative and quantitative articles were reviewed in this paper. The qualitative articles provide insights into problems by helping to understand the reason and opinions. The quantitative articles use measurable data to express facts and discover research patterns.

## SOURCES OF DHA IN MILK

DHA in milk comes from three major sources: those synthesized from endogenous ALA, those synthesized by the microorganisms in the rumen and intestines of cows, and those converted from the diet.

### Metabolic conversion of ALA to DHA

DHA can be synthesized from ALA through metabolic pathways in the liver (Fig. 1) (*Kabeya et al., 2018*; *Kim et al., 2014*). The process occurs in the endoplasmic reticulum and peroxisome, which are organelles. ALA is desaturated to stearidonic acid in the endoplasmic reticulum (C18:4n-3) catalyzed in a rate-limiting reaction by Δ6 desaturase and then converted to tetracosahexaenoic acid (C24:6n-3). Tetracosahexaenoic acid is then transferred into peroxisome where it undergoes $\beta$-oxidation to form DHA. The reaction sequence for converting ALA to eicosatetraenoic acid (C20:4n-3) is as follows: ALA → eicosatrienoic acid (C20:3n-3) → C20:4n-3 (*Kabeya et al., 2018*). Δ6 and Δ5 desaturases are the key enzymes in the metabolic pathways (*Missotten et al., 2009*) and the activity of these two desaturate enzymes can determine the amount of DHA synthesized. For example, the expressions of Δ6 and Δ5 desaturase in human subjects are positively correlated with SFA and PUFA but negatively correlated with linoleic acid (LA) and ALA in foods (*Xiang et al., 2006*). Omega-6 polyunsaturated fatty acids (n-6 PUFA) are essential in human and animal diets (*Saini & Keum, 2018*). Changing the ratio of n-3:n-6 PUFA in the diet can influence the expression of Δ6 desaturase enzyme in rats (*Missotten et al., 2009*; *Neuringer, Anderson & Connor, 1988*). *Missotten et al. (2009)* added fish oil and linseed oil to pig diets and determined the expressions of Δ6 and Δ5 desaturase in the liver, subcutaneous fat, and the *longissimus dorsi* muscle. The addition of fish oil increased the expression of Δ5 desaturase only in the *longissimus dorsi* muscle, but not in the liver or subcutaneous fat; the addition of linseed oil had no effect on the expression of Δ5 desaturase in all three tissues; the expression of Δ6 desaturase in all three tissues was not affected by either type of oil (*Missotten et al., 2009*). Studies in rats have shown that protein (*Narce et al., 1988*) and micromineral (*Johnson et al., 1989*) depletion in the diet reduced the activity of the Δ6 desaturase enzyme. There have been no reports on the effect of dietary fat on the expression of Δ6 and Δ5 desaturases in cows.

### Synthesis of DHA by rumen microorganisms

Microorganisms, including microbes in the ocean and environment, are able to synthesize DHA *de novo* via a polyketide synthase pathway (*Kabeya et al., 2018*; *Dongming, Jackson & Quinn, 2015*; *Xue et al., 2013*). Diverse and interdependent populations of bacteria, protozoa, and fungi inhabit the rumen of dairy cattle (*Russell & Rychlik, 2001*) but no studies have been conducted to date that show the relationship between rumen microbes and DHA metabolism. Some articles have indicated that microbes in the rumen may synthesize DHA independently. For example, *Bianchi et al. (2014)* found that *Cellulophaga* can produce DHA. *Cellulophaga* belongs to phylum Bacteroidetes which is the most abundant phylum in the rumen of older dairy cows (*Jami et al., 2013*), and infers that there may be a kind of microbe (belong to phylum Bacteroidetes) in the rumen that can produce

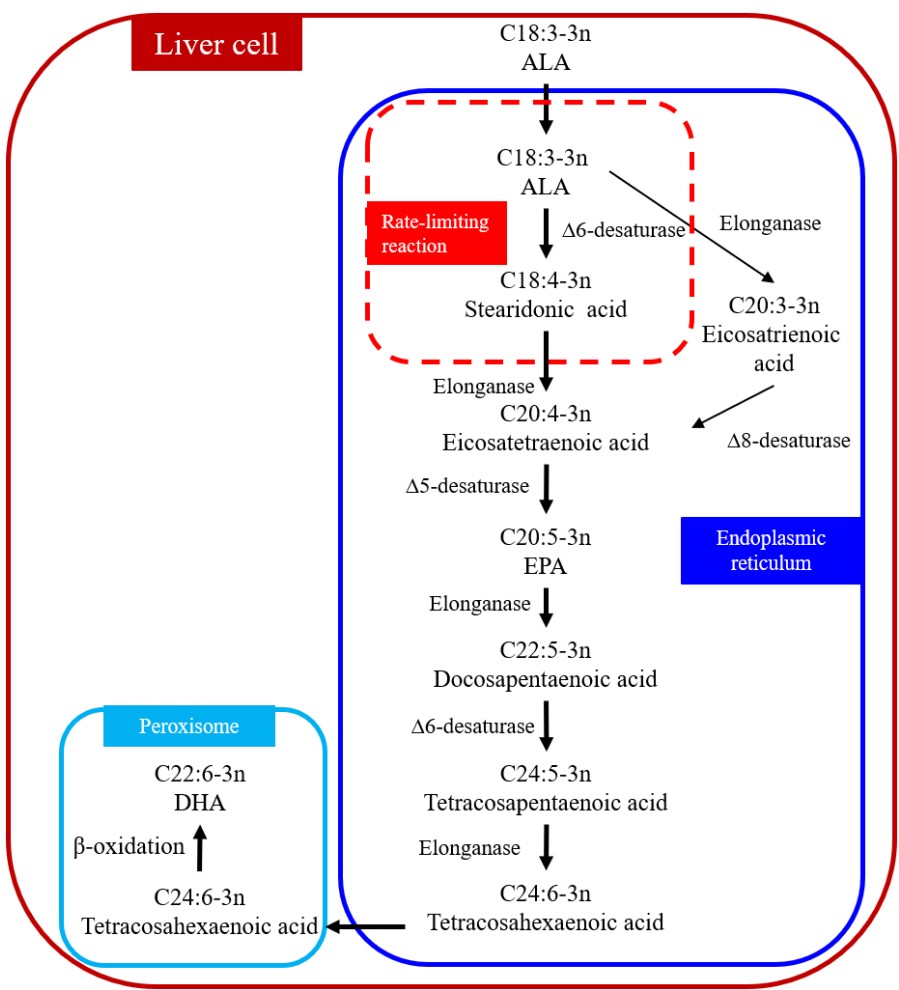

**Figure 1** **Biosynthetic conversion pathway of ALA to DHA.** Abbreviations: ALA, $\alpha$-linolenic acid; EPA, eicosapentaenoic acid; DHA, docosahexaenoic acid. The first step that ALA is converted to stearidonic acid (C18:4-3n) is a rate-limiting reaction. Data is taken from *Kabeya et al. (2018)* and *Kim et al. (2014)*.

DHA. *Yarrowia lipolytica* is a yeast widely distributed in the natural environment that can also produce DHA (*Dongming, Jackson & Quinn, 2015*; *Damude et al., 2006*; *Gong et al., 2014*), however, it is not clear if yeast living in the rumen can produce DHA (*Prakasan et al., 2013*). In an in vivo experiment with dairy cows, the animals were fed diets containing linseeds or no linseeds, and no DHA at all, but DHA was subsequently found in the duodenal chyme (0.07 g/d and 0.08 g/d, respectively) (*Shingfield et al., 2011*; *Kairenius, Toivonen & Shingfield, 2011*). The DHA in the chyme came from the rumen and originated from synthesis by rumen microorganisms or from the blood circulation into salivary secretions.

Additional research may need to focus on the identification of DHA-producing bacteria in the rumen. Increasing the content of DHA-producing bacteria present may increase DHA synthesis in the rumen, which results in more DHA in milk. In addition, there may

exist interrelationships in DHA synthesis between microbial species with a functional network (*Moraïs & Mizrahi, 2019*), which should be explored.

## Dietary DHA

An increase in dietary DHA can significantly increase the bodily content of DHA and the subsequent milk content of DHA (*Scollan et al., 2001a*; *Vahmani, Fredeen & Glover, 2013*). Currently, the major dietary DHA sources for cows are fish oil and microalgae products.

## Fish oil

Fish oils contain a variety of n-3 PUFA, of which EPA and DHA are the most abundant (*Mahla et al., 2017*). Studies have been conducted on supplementing the diets of dairy cattle with fish oil (Table 1) and most studies indicate that this practice could reduce milk fat. The study conducted by *Pirondini et al. (2015)* showed no negative effects of fish oil (0.8% dry matter) on milk fat when cattle were provided a low starch diet. The type of diet fed, including the percentage of forage (*Shingfield et al., 2003*) and type of forage (*Chilliard, Ferlay & Doreau, 2001*), plays an important role in milk fat concentrations.

The supplementation of fish oil alone or fish oil combined with other oils (such as extruded soybean, canola oil) all resulted in improved DHA concentrations in milk (*Vahmani, Fredeen & Glover, 2013*; *AbuGhazaleh et al., 2002*; *Ramaswamy et al., 2001*; *Vafa et al., 2012*; *Whitlock et al., 2002*). *Kairenius et al. (2015)*, supplemented with fish oil at doses of 75, 150 and 300g/day (around 0.4, 0.8 and 1.88% diet) which increased the DHA concentration in milk (0.03, 0.05 and 0.10 g/100g total milk fatty acid or 0.22, 0.39 and 0.67 g/day in milk). Other studies have shown a positive correlation for DHA content between dietary intake and milk concentrations (*Lacasse et al., 2002*). However, increased supplementation had no constant linear relationship between dietary DHA intakes and DHA concentrations in milk. *Donovan et al. (2000)* showed that supplementation with fish oil at 0, 1 and 2% of total diet increased DHA concentrations in milk, but that the concentration decreased with 3% of total diet fish oil supplementation. *Kairenius et al. (2015)* reported no difference between the control group and 75 g/day supplementation group for DHA concentrations (0.03 to 0.03 g/100g total milk fatty acid or 0.22 to 0.22 g/day in milk). There may be a liner relationship between fish oil supplementation and milk DHA within certain range, which may be between 0.4% to 3%.

The DHA content of milk is also affected by the host's metabolism and the biohydrogenation pathway in the rumen. In theory, minimizing the effects of rumen biohydrogenation in the rumen could increase the DHA content of milk (*Casta Eda-Gutiérrez et al., 2007*). However, *Lacasse et al. (2002)* reported that supplementation with fish oil or rumen-protected fish oil at the same doses in the diet made no difference in the DHA concentration of milk. This effect may be due to the reduced digestibility of DHA in rumen-protected fish oil. Dietary supplementation of fish oil can increase the DHA content of milk, but the effect of DHA intake is affected by many factors that need to be quantitatively defined.

**Table 1 The effects of dietary supplementation of fish oil on milk fat content.**

| Treatment | Fish oil supplement | Diet DHA intake (g/d) | Milk fatty acid content (%) | DHA content in milk (g/d) | Increase of DHA content in milk (compared with control group) (g/d) | Reference |
|---|---|---|---|---|---|---|
| C=basal diet | – | – | 3.52 | 0.26 | – | *Vahmani, Fredeen & Glover (2013)* |
| T=basal diet + RUFO | 200 g/d | 24.7 | 3.37 | 1.68 | 1.42 | |
| Ca=basal diet | – | 0.00 | 4.30 | 0.00 | – | *Pirondini et al. (2015)* |
| Ta=basal diet + RUFO | 0.80% | 17.6 | 4.51 | 0.40 | 0.40 | |
| C=basal diet | – | 0.21 | 3.46 | 0.40 | – | *AbuGhazaleh et al. (2002)* |
| T1=basal diet +RUFO | 2.00% | 21.3 | 3.22 | 2.49 | 2.09 | |
| T2=basal diet +RUFO | 1.00% | 12.2 | 3.45 | 1.46 | 1.06 | |
| C = basal diet | – | ND | 3.28 | ND | – | *Ramaswamy et al. (2001)* |
| T = basal diet + RUFO | 2.00% | ND | 2.56 | ND | – | |
| C=basal diet | – | 0.35 | 3.40 | 0.47 | – | *Vafa et al. (2012)* |
| T1=basal diet + RUFO | 2.00% | 31.7 | 2.30 | 2.18 | 1.71 | |
| T2=basal diet + RUFO | 1.00% | 12.9 | 2.45 | 0.84 | 0.37 | |
| Ca=basal diet | – | – | 3.48 | 0.56 | – | *Whitlock et al. (2002)* |
| Ta1=basal diet +RUFO | 2.00% | 11.1 | 2.87 | 1.59 | 1.03 | |
| Ta2=basal diet +RUFO | 1.00% | 9.07 | 3.11 | 0.97 | 0.41 | |
| Cb = basal diet | – | – | 4.36 | 0.12 | – | |
| Tb = basal diet + RUFO | 0.80% | 18.5 | 3.87 | 0.34 | 0.22 | |
| C = basal diet | – | ND | 3.27 | 0.30 | – | *Kairenius et al. (2015)* |
| T1 = basal diet + RUFO | 75 g/d | ND | 3.23 | 0.28 | −0.02 | |
| T2 = basal diet + RUFO | 150 g/d | ND | 3.14 | 0.42 | 0.12 | |
| T3 = basal diet + RUFO | 300 g/d | ND | 3.33 | 0.59 | 0.29 | |
| C = basal diet | – | ND | ND | 0.30 | – | *Lacasse et al. (2002)* |
| T1 = basal diet + RUFO | 3.70% | ND | ND | 1.03 | 0.73 | |
| T2 = basal diet + RPFO | 1.80% | ND | ND | 0.89 | 0.59 | |
| T3 = basal diet + RPFO | 3.70% | ND | ND | 1.04 | 0.74 | |
| C=basal diet | – | – | 2.94 | 0.19 | – | *Donovan et al. (2000)* |
| T1=basal diet +RUFO | 1.00% | 14.30 | 2.77 | 0.57 | 0.38 | |
| T2=basal diet +RUFO | 2.00% | 50.66 | 2.35 | 1.97 | 1.78 | |
| T3=basal diet +RUFO | 3.00% | 92.63 | 2.28 | 1.25 | 1.06 | |
| C = basal diet | – | ND | 3.34 | ND | – | *Baer et al. (2001)* |
| T = basal diet + RUFO | 2.00% | ND | 2.27 | ND | – | |
| C = basal diet | – | ND | 4.56 | 1 | – | *Shingfield et al. (2006)* |
| T = basal diet + RUFO | 1.50% | ND | 2.87 | 0.54 | 0.54 | |

**Notes.**
Fatty acid ≈ triacylglycerols + diacylglycerols + monoacylglycerols + free fatty acids.
Milk fatty acid = milk fat content × 99.13% (*MacGibbon & Taylor, 2006*).
C, control; Ca,b, controls in article; Ta,b, treatments in article; T1, 2, 3, treatments.
RPFO, rumen protected fish oil; RUFO, rumen unprotected fish oil; ND, Not detected.

**Table 2  The effects of dietary supplementation of microalgae on milk fat content.**

| Treatment | Microalgae supplement (g/d) | Diet DHA intake (g/d) | Milk fatty acid (%) | DHA content in milk g/d | Increase of DHA content in milk (compared with control group) (g/d) | Reference |
|---|---|---|---|---|---|---|
| C=basal diet | – | – | 4.75 | 0.13 | – | *Boeckaert et al. (2008)* |
| T=basal diet + RPA | 899 | 43.7 | 2.18 | 1.45 | 1.32 | |
| C=basal diet | – | 0.05 | 3.36 | 0.70 | – | *Fougère, Delavaud & Bernard (2018)* |
| T=basal diet + RUA | 310 | 115 | 2.62 | 13.6 | 12.9 | |
| C=basal diet | – | ND | 4.75 | 0.10 | – | *Póti et al. (2015)* |
| T=basal diet + RUA | 150 | ND | 3.46 | 0.14 | 0.04 | |
| C=basal diet | – | – | 3.67 | 0.00 | – | *Franklin et al. (1999)* |
| T1=basal diet + RPA | 910 | 29.2 | 2.92 | 5.15 | 5.15 | |
| T2=basal diet + RUA | 910 | 35.9 | 2.92 | 3.23 | 3.23 | |
| C=basal diet | – | – | 3.47 | 0.10 | – | *Stamey et al. (2012)* |
| T1= basal diet + 0.5 × RUA | 150 | 21.6 | 3.97 | 0.50 | 0.40 | |
| T2= basal diet + 1 × RUA | 300 | 43.2 | 3.27 | 0.59 | 0.49 | |
| T3= basal diet + 1 × RUA oil | 194 | 27.4 | 3.27 | 0.30 | 0.20 | |
| C=basal diet | – | – | 4.93 | 0.44 | – | *Moate et al. (2013)* |
| T1=basal diet + RUA | 125 | 25.0 | 3.75 | 3.51 | 3.07 | |
| T2=basal diet + RUA | 250 | 50.0 | 3.67 | 5.09 | 4.65 | |
| T3=basal diet + RUA | 375 | 75.0 | 3.80 | 7.70 | 7.26 | |
| C=basal (diet + Hydrogenated palm oil fat | – | – | 3.90 | – | – | *Moran et al. (2018)* |
| T=basal diet + RUA | 100 | 17.8 | 3.81 | 1.32 | 1.32 | |

**Notes.**

Fatty acid ≈ triacylglycerols + diacylglycerols + monoacylglycerols + free fatty acids.

Milk fatty acid = milk fat content × 99.13% (*MacGibbon & Taylor, 2006*).

C, control; T1, 2, 3, treatments.

RPA, rumen protected algae; RUA, rumen unprotected algae; ND, Not detected.

## Microalgae

Microalgae are microscopic photosynthetic organisms found in marine and fresh waters that are used as an animal feed (*Priyadarshani & Rath, 2012*). Microalgae are a good source of protein, carbohydrates, and long chain PUFA, some of which are rich in DHA (*Ryckebosch et al., 2014*). Microalgae have been shown to improve the DHA content in milk when used as an additive to dairy cattle feed (*Altomonte et al., 2018*). The effects of microalgae supplementation on the fatty acid profile of milk are summarized in Table 2.

Supplementation with microalgae has been shown to improve the DHA concentration of milk with a negative effect on the overall fat content of milk. Microalgae supplementation has a liner relationship with the DHA content of milk (*Altomonte et al., 2018*; *Boeckaert et al., 2008*; *Fougère, Delavaud & Bernard, 2018*; *Póti et al., 2015*) and fish oil has been shown to have the same effect. Three microalgae feeding styles were utilized (microalgae,

rumen protect microalgae, and microalgae oil) and each produced unique results. The feeding of rumen-protected microalgae can improve the concentration of milk DHA markedly, compared to feeding microalgae alone (*Franklin et al., 1999*). Rumen-protected microalgae can reduce the biohydrogenation of DHA in the rumen. *Stamey et al. (2012)* supplemented with 150 g/day of microalgae and 194 g/day of microalgae oil, respectively, and found that the microalgae oil supplementation produced a lower milk fat content, DHA concentration and efficient transport of dietary DHA into milk compared with supplementation of microalgae. The DHA in microalgae oil is able to be biohydrogenated in the rumen more easily than microalgae, although DHA is chosen as a source of dietary DHA more often. *Moate et al. (2013)* reported that there is an exact linear relationship between microalgae intake and the DHA content of milk. However, no experiments have revealed the range in which microalgae supplementation has a linear relationship with the DHA concentration of milk.

An increased concentration of DHA in milk depends on the DHA content in dietary microalgae and is dependent on the species of microalgae and its processing methods. There are many kinds of microalgae that can be used in animal diets with substantially different levels of DHA (*Madeira et al., 2017*) that can be influenced by the way they are processed. Protecting microalgae from rumen degradation can preserve approximately 45% of the DHA content versus un-protected microalgae (*Stamey et al., 2012*).

### Transport ratio of DHA in milk

The efficiency of DHA incorporation from the feed into milk was low, as showed in Table 3. The incorporation efficiency of DHA can be calculated as the ratio of milk DHA content to dietary DHA intake. Fish oil supplementation increased the DHA content in milk by approximately 6.86% (range from 1.35% to 14.4%), while microalgae supplementation increased it approximately 7.08% (range from 1.09% to 14.0%). The efficiency can be influenced by many factors.

## TRANSPORTATION OF DIETARY DHA INTO MILK

Figure 2 shows how dietary DHA is moved through the body into the milk. The majority of dietary DHA is hydrogenated in the rumen, with 60–98% of DHA transformed in the rumen to the corresponding geometric isomers via cis-trans isomerization of double bonds in DHA (*NRC, 2001*; *Scollan et al., 2001a*; *Scollan et al., 2001b*; *Kim et al., 2008*; *Kim et al., 2008*; *Shingfield et al., 2011*; *Shingfield et al., 2012*; *Kairenius et al., 2018*). Intact DHA (or by-pass) flows into the small intestine where approximately 70–100% is absorbed (*Doreau & Ferlay, 1994*; *Wachira et al., 2000*; *NRC, 2001*; *Scollan et al., 2001a*; *Scollan et al., 2001b*; *Mattos et al., 2004*). It is then absorbed via the lymphatic system into the blood circulation system (*Doreau & Ferlay, 1994*; *Scollan et al., 2001b*; *NRC, 2001*; *Wachira et al., 2000*) and is transported via the blood into various tissues and organs of the body, including the brain (*Al-Ghannami, Al-Adawi & Ghebremeskel, 2019*), bones (*Saini & Keum, 2018*), and the reproductive system (*Gholami et al., 2010*), where it is used for tissue repair or energy supply via the $\beta$-oxidative pathway. Only 13–25% of DHA absorbed from the small

**Table 3  Efficiency of dietary incorporation of DHA into milk.**

| DHA source | DHA in take (g/d) | Milk DHA yield (g/d) | Efficiency (%) | Reference |
|---|---|---|---|---|
| Fish oil | 14.30 | 0.57 | 3.99 | *Donovan et al. (2000)* |
| Fish oil | 50.66 | 1.97 | 3.89 | *Donovan et al. (2000)* |
| Fish oil | 92.63 | 1.25 | 1.35 | *Donovan et al. (2000)* |
| Fish oil | 21.27 | 2.49 | 11.7 | *AbuGhazaleh et al. (2002)* |
| Fish oil | 12.20 | 1.46 | 12.0 | *AbuGhazaleh et al. (2002)* |
| Fish oil | 11.07 | 1.59 | 14.4 | *Whitlock et al. (2002)* |
| Fish oil | 9.07 | 0.97 | 10.7 | *Whitlock et al. (2002)* |
| Fish oil | 31.71 | 2.18 | 6.87 | *Vafa et al. (2012)* |
| Fish oil | 12.87 | 0.84 | 6.53 | *Vafa et al. (2012)* |
| Fish oil | 24.66 | 1.68 | 6.81 | *Vahmani, Fredeen & Glover (2013)* |
| Fish oil | 17.55 | 0.40 | 2.28 | *Pirondini et al. (2015)* |
| Fish oil | 18.53 | 0.34 | 1.83 | *Pirondini et al. (2015)* |
| Microalgae | 35.94 | 3.23 | 8.99 | *Franklin et al. (1999)* |
| Microalgae | 43.68 | 1.45 | 3.32 | *Boeckaert et al. (2008)* |
| Microalgae | 25.00 | 3.51 | 14.0 | *Moate et al. (2013)* |
| Microalgae | 50.00 | 5.09 | 10.2 | *Moate et al. (2013)* |
| Microalgae | 75.00 | 7.70 | 10.3 | *Moate et al. (2013)* |
| Microalgae | 21.61 | 0.50 | 2.31 | *Stamey et al. (2012)* |
| Microalgae | 43.23 | 0.59 | 1.36 | *Stamey et al. (2012)* |
| Microalgae oil | 27.41 | 0.30 | 1.09 | *Stamey et al. (2012)* |
| Microalgae | 17.82 | 1.32 | 7.41 | *Moran et al. (2018)* |
| Microalgae | 115.5 | 13.6 | 11.8 | *Fougère, Delavaud & Bernard (2018)* |

**Notes.**
DHA intake: reported in the article or calculation: dry matter intake × fatty content of dietDHA content.
Fatty acid ≈ triacylglycerols + diacylglycerols + monoacylglycerols + free fatty acids.
Milk fatty acid = milk fat content × 99.13% (*MacGibbon & Taylor, 2006*).
Milk DHA yield: reported in the article or calculation: milk fatty yieldDHA content.
Efficiency: milk DHA/diet DHA.

intestine is transported into milk through the mammary gland cells (*Shingfield, Bonnet & Scollan, 2013*).

The biohydrogenation of DHA in the rumen increases the amount of DHA lost by the body (Fig. 2) (*Kairenius et al., 2018*; *Kim et al., 2008*; *Mattos et al., 2004*; *Scollan et al., 2001a*; *Scollan et al., 2001b*; *Shingfield et al., 2003*; *Shingfield et al., 2012*; *Shingfield et al., 2011*; *NRC, 2001*; *Wachira et al., 2000*). Therefore, reducing the biohydrogenation of DHA in the rumen is important for improving the DHA concentration of milk.

## Ruminal Biohydrogenation

Ruminal biohydrogenation limits the transportation of dietary DHA into milk and is influenced by rumen microbes. Rumen microorganisms include bacteria, protozoa, and fungi. Bacteria play an important role in the biohydrogenation process (*Louren, Ramos-Morales & Wallace, 2010*). DHA has two dietary forms: free fatty acids and triacylglycerols. Triacylglycerols must be converted into free fatty acids and then must

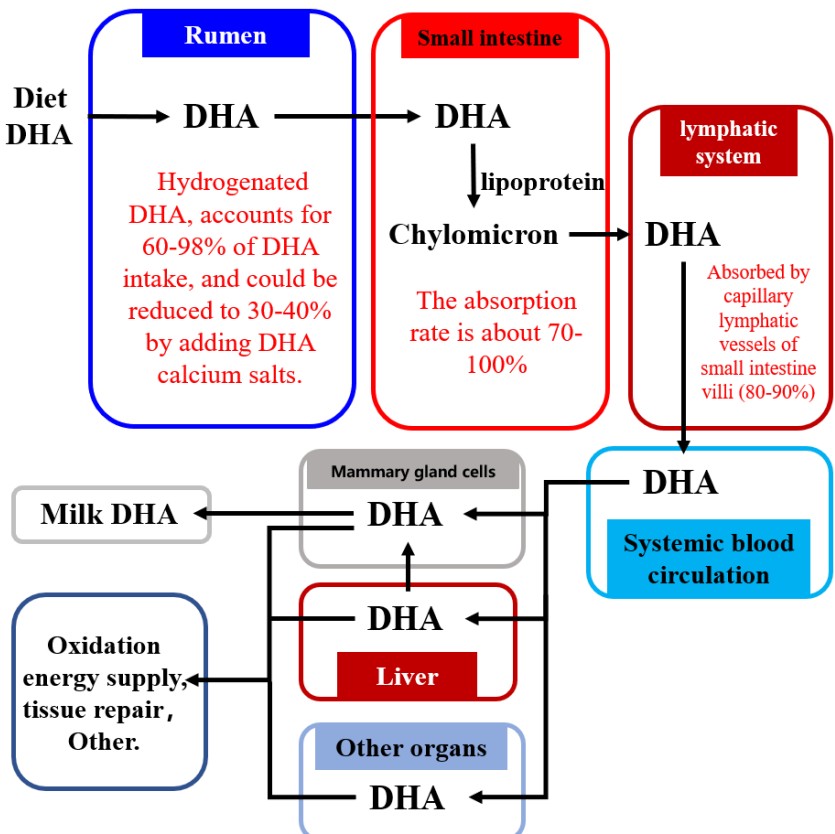

**Figure 2** **The pathway of DHA transportation into milk.** Abbreviations: DHA, docosahexaenoic acid. Data is taken from *Shingfield et al. (2003)*, *NRC (2001)*, *Wachira et al. (2000)*, and *Mattos et al. (2004)*.

undergo biohydrogenation by the rumen microbes. Thus, two kinds of microorganisms exist with lipolytic effects and biohydrogenation properties.

The lipolytic effect of triacylglycerols mainly depended on the lipase, *Anaerovibrio lipolyticus,* which is a prominent ruminal lipase-producing bacterium (*Hungate, 1966*). Three putative lipase genes were identified from the draft genome of *Anaerovibrio lipolyticus* (alipA, alipB, alipC) (*Privé et al., 2013*) and had greater hydrolytic activity against caprylate (C8:0), laurate (C12:0), and myristate (C14:0). *Butyrivibrio fibrisolvens*, *Propionibacterium* (*Edwards et al., 2012*) *Clostridium*, *Propionibacterium*, *Staphylococcus* (*Edwards et al., 2013*), and *Pseudomonas aeruginosa* (*Priji et al., 2017*) are among the bacteria that have the ability to decompose triacylglycerols. *Sargolzehi et al. (2015)* showed that pyridostigmine bromide could decrease the lipase activity and the immunization against lipase may also inhibit the decomposition of triacylglycerols, just like the immunization against rumen urease inhibits ureolysis in the rumen (*Zhao et al., 2015*).

*Butyrivibrio sp.* is a genus of an important microbe that hydrogenates PUFA in the rumen; it includes *Butyrivibrio fibrisolvens (B. fibrisolvens)*, and *Butyrivibrio proteoclasticus (B. proteoclasticus)*. *B. fibrisolvens* can produce isomerase and change the PUFA structure (for example, converting LA into CLA) (*Kepler et al., 1966*). Trans-11 vaccenic acid (C18:1),

converted from LA, can be hydrogenated to stearic acid (C18:0) by *B. proteoclasticus* (*Jenkins et al., 2007*). However, some studies show that *B. fibrisolvens* failed to successfully induce DHA hydrogenation in the rumen (*Jeyanathan et al., 2016*; *Maia et al., 2007*). *B. proteoclasticus* could hydrogenate DHA (*Jeyanathan et al., 2016*) in vitro in a growth medium containing autoclaved ruminal fluid. Bacterial species, such as *Acetobacter* (*Bainbridge et al., 2016*) and *Bacillus* (*Petri et al., 2014*), but not *Butyrivibrio sp.*, can affect DHA biohydrogenation. However, an experiment by *Sakurama et al. (2014)* reported that no bacteria (100 strains of anaerobic bacteria were used, *Acetobacter* was included) metabolized DHA. Dietary PUFA has been shown to strongly influence microbial profiles in the rumen. Many studies have shown that DHA intake in a reduction of *B. fibrisolvens* in the rumen in a dose-dependent manner (*Shingfield et al., 2012*; *Maia et al., 2010*; *Shinji et al., 2009*). *Abughazaleh & Ishlak (2014)* reported that supplement with DHA could reduce the abundance of *B. proteoclasticus*, but other experiments have shown no effect. *Shingfield et al. (2012)* proposed that DHA and other unsaturated fatty acids could lengthen the bacteria's lag phase. It is well known that rumen bacteria release hydrogens and secrete isomerases, which may hydrogenate the double bonds in unsaturated fatty acids. The biohydrogenase in the rumen is a major factor regulating the biohydrogenation of PUFA. Further studies should focus on the relationship between rumen microbes, DHA, and biohydrogenase.

The formation of DHA can also be influenced by the intake of LA. LA can be converted into highly unsaturated fatty acids (HUFA) in vivo. LA and ALA share the same family of enzymes in the formation of HUFA (*Fleming & Kris-Etherton, 2014*), and compete with one another for enzyme uptake (*Gibson, Muhlhausler & Makrides, 2015*). Increasing the intake of LA may reduce the formation of DHA from ALA. A study showed that high LA diet could reduce the content of DHA in milk (*Aprianita et al., 2014*). However, DHA synthesis by ALA in tissues is very low and yet not reported in dairy cows. Ruminal biohydrogenation process is well-studied and understood. The content of unsaturated fatty acids in the diet can influence the biohydrogenation of DHA in the rumen, to various effects. In an in vitro study, *Chow et al. (2004)* found that adding LA and ALA could reduce the biohydrogenation of DHA, which was confirmed in an in vitro experiment (*Wasowska et al., 2006*). *Shingfield et al. (2011)* found that dietary supplementation of both fish oil and linseed oil at a ratio of 1:1 reduced the hydrogenation of DHA, but increased the hydrogenation of ALA in the rumen. However, *Kairenius et al. (2018)* reported that dietary addition of linseed oil or sunflower seed oil could promote the biohydrogenation of DHA, EPA, and ALA compared with fish oil alone. We determined that biohydrogenase is not fatty-acid specific and competition exists among unsaturated fatty acids. Short-chain unsaturated fatty acids may tend to be biohydrogenated more readily than long-chain PUFA. Therefore, understanding the mechanisms of biohydrogenation for unsaturated fatty acids and the interactions among these fatty acids in the rumen will help develop dietary strategies to reduce DHA biohydrogenation.

## Biohydrogenation pathways of DHA in rumen

Studies on the DHA biohydrogenation pathway in the rumen are limited. In 2007, *Jenkins et al. (2007)* speculated that the first step in the process of DHA biohydrogenation is to convert DHA to a C22:6 isomer that is then hydrogenated to C22:5 fatty acid. However, *Kairenius, Toivonen & Shingfield (2011)* showed that the C22:6 isomer was not detectable in the DHA hydrogenation process, likely due to its short lifetime or the limitation of the analytic method (*Escobar et al., 2016*). *Aldai et al. (2018)* investigated the biohydrogenation process of DHA in in vitro fermentation using sheep rumen fluid as the inoculator, and determined the metabolites of DHA at 0, 1-, 2-, 3-, and 6- hours after fermentation. They found that DHA was initially transformed into mono *trans* methylene interrupted DHA and monoconjugated DHA. Nevertheless, the DHA hydrogenation process started from the isomer formation.

*Jeyanathan et al. (2016)* used in vitro anaerobic fermentation with a single strain of *Butyrivibrio proteoclasticus* P18 to explore the biohydrogenation process of DHA during fermentation and showed the product in the DHA biohydrogenation pathway. This experiment showed that 12 kinds of DHA intermediates (C22:5, C22:4, C22:3 and C22:2 isomers) were transformed in 48 h. *Toral et al. (2018)* found that Docosapentaenoic acid may be a major DHA intermediate product. DHA intermediates and the hydrogenation pathway in the rumen are illustrated in Fig. 3.

The literature is lacking for enzymes that may regulate the DHA hydrogenation pathway in the rumen. According to a report by *Toral et al. (2018)*, some enzymes may exist relating to hydrogenation, isomerization, and migration in the EPA hydrogenation pathway. However, the specific enzymes have not been identified yet, and further studies should focus on the enzymes that regulate the DHA hydrogenation pathway in the rumen.

## OTHER FACTOR

### *Fat-1* gene modification

The Fat-1 gene is present in *Caenorhabditis elegans*, a free-living nematode. *Spychalla, Kinney & Browse (2007)* first reported the *fat-1* gene in *Caenorhabditis elegans*, and specifically expressed the gene in Arabidopsis, confirming the cDNA (complementary DNA) sequence of the *fat-1* gene. The translation product of the *fat-1* gene is n-3 PUFA dehydrogenase, which can catalyze the formation of the corresponding n-3 PUFA using 18-20 carbon n-6 PUFA as the substrate (*Kang, 2005*). The expression of the *fat-1* gene can promote the synthesis of n-3 PUFA in nematodes. *Liu et al. (2017)* constructed the eukaryotic expression vector *pef-gfp-fat-1*, then transfected *pef-gfp-fat-1* into cow fetal fibroblast cells and determined the fatty acid profile. They found that the expression of *fat-1* gene could increase the DHA concentration in the cells. The birth of transgenic cows that carried and expressed the mammalianized *fat-1* gene (*mfat-1*) (*Wu et al., 2012*) and a transgenic cow showed increased n-3 PUFA profiles and reduced n-6 PUFA in their tissues and milk (*Liu et al., 2017*; *Wu et al., 2012*). The effect of the *fat-1* gene on the conversion of n-6 to n-3 PUFA was also confirmed in a transgenic-pig model (*Kang et al., 2004*; *Li et al., 2018*). These findings need to be validated in a large cohort of transgenic animals to support these conclusions.

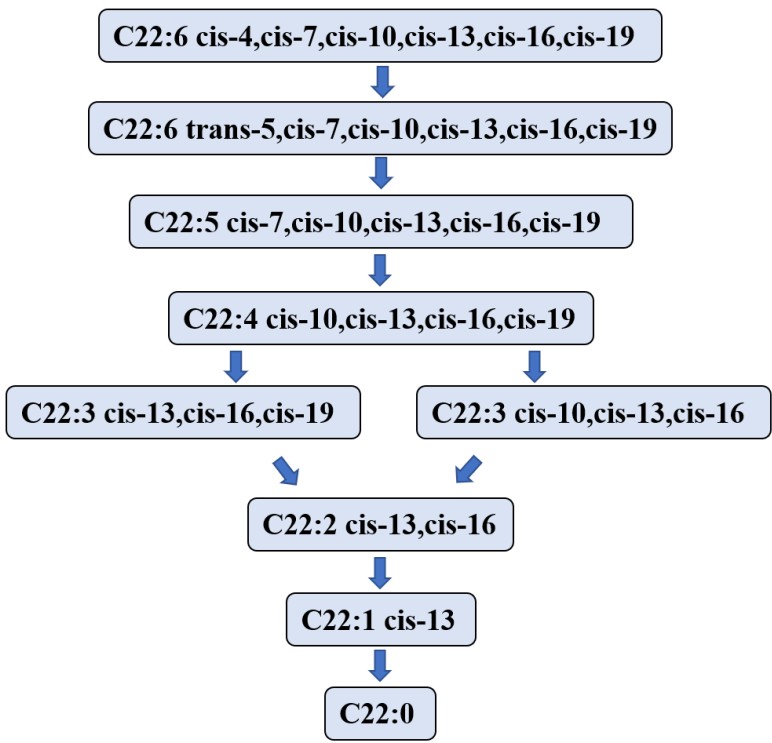

**Figure 3 The putative biohydrogenation pathway of DHA in the rumen.** Abbreviations: DHA, docosahexaenoic acid, C22:6 cis-4,cis-7,cis-10,cis-13,cis-16,cis-19. The arrows represent possible major pathways. Neither all putative fatty acids nor the numerous interconversions among C22:6 isomers are presented. Data is taken from *Kairenius et al. (2018)*, *Shingfield et al. (2012)*, *Jenkins et al. (2007)*, *Aldai et al. (2018)*, and *Jeyanathan et al. (2016)*.

## CONCLUSIONS

The literature is limited regarding the conversion of ALA to DHA in tissues and its effect on DHA content in milk. Many studies have focused on increasing the DHA concentration in milk by manipulation of the DHA supply in the diet. Many dietary factors can influence DHA's passage into milk and their effects need to be quantitated. The majority of dietary DHA is biohydrogenated in the rumen. It is extremely important to reduce our reliance on rumen biohydrogenation and find alternative means for synthesizing DHA.

The *fat-1* gene from nematodes is highly effective in converting n-6 PUFA to n-3 PUFA. Since the gene does not exist in mammals, transgenic techniques have been applied, which have been successful in cows, pigs and mice. Thus, it may be worthwhile to examine enlarging the transgenic population.

## ACKNOWLEDGEMENTS

We thank Professor Shimin Liu from the University of Western Australia for their advice.

### Funding

This study was supported by the National Natural Science Foundation of China (grant number: 31601963), The Agricultural Science and Technology Innovation Program (ASTIP-IAS12) and Modern Agro-Industry Technology Research System of the PR China (CARS-36), The Scientific Research Project for Major Achievements of The Agricultural Science and Technology Innovation Program (CAAS-ZDXT2019004). The funders had no role in study design, data collection and analysis, decision to publish, or preparation of the manuscript.

### Grant Disclosures

The following grant information was disclosed by the authors:
National Natural Science Foundation of China: 31601963.
The Agricultural Science and Technology Innovation Program: ASTIP-IAS12.
Modern Agro-Industry Technology Research System of the PR China: CARS-36.
The Scientific Research Project for Major Achievements of The Agricultural Science and Technology Innovation Program: CAAS-ZDXT2019004.

### Competing Interests

We declare that we have no financial and personal relationships with other people or organizations that can inappropriately influence our work. There is no professional or other personal interest of any nature or kind in any product, service and/or company that could be construed as influencing the content of this paper. Xueyin Qu is employed by Tianjin Mengde Groups Co., Ltd. Our lab collaborates with Tianjin Mengde Groups Co., Ltd.

### Author Contributions

- Guoxin Huang and Yangdong Zhang conceived and designed the experiments, analyzed the data, prepared figures and/or tables, and approved the final draft.
- Qingbiao Xu conceived and designed the experiments, prepared figures and/or tables, and approved the final draft.
- Nan Zheng, Shengguo Zhao, Kaizhen Liu, Xueyin Qu and Jing Yu performed the experiments, authored or reviewed drafts of the paper, and approved the final draft.
- Jiaqi Wang conceived and designed the experiments, prepared figures and/or tables, and approved the final draft.

### Data Availability

This is a review article; there is no raw data or code.

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
