# Peer review of "DHA content in milk and biohydrogenation pathway in rumen: a review"

_PeerJ, doi:10.7717/peerj.10230_

## Round 0.1 · original submission · Major Revisions

The manuscript subject is quite relevant to the journal and of wider scientific interest. However it needs to be corrected, keeping in mind reviewer's comments. Specifically, language needs some help, overall structure should be properly rearranged, need to add some missing (important) information, and more. Please read carefully reviewer's comments (one reviewer also attached additional comments as an annotated attachment), and incorporate those changes in the manuscript.

Reviewer 1 ·

Basic reporting

The Manuscript 48138v1 is a review on DHA content in milk and biohydrogenation pathway in rumen. The manuscript is a review but it is difficult to understand exactly what is the objective of the review? It is difficult to find a reflection or input from the authors in each of their results and discussion paragraphs.
The review lack of structure and it is difficult to understand its rationale. For example, suddenly Fat-1 gene appears in the end of the document, but this is not explained in the introduction? It just seems that authors just enumerated results from different documents but they do not discussed them with profound criticism. The manuscript should be revised for its English style.

Experimental design

Line 72 should say Methods or Methodology
Lines 73-79 provide range of dates or months you used for your search, provide data bases used i.e., scopus, web of knowledge, google scholar?
What do you mean by qualitative and quantitative articles?

Validity of the findings

Line 72 should say Methods or Methodology
Lines 73-79 provide range of dates or months you used for your search, provide data bases used i.e., scopus, web of knowledge, google scholar?
What do you mean by qualitative and quantitative articles?

Additional comments

The Manuscript 48138v1 is a review on DHA content in milk and biohydrogenation pathway in rumen. The manuscript is a review but it is difficult to understand exactly what is the objective of the review? It is difficult to find a reflection or input from the authors in each of their results and discussion paragraphs.
The review lack of structure and it is difficult to understand its rationale. For example, suddenly Fat-1 gene appears in the end of the document, but this is not explained in the introduction? It just seems that authors just enumerated results from different documents but they do not discussed them with profound criticism. The manuscript should be revised for its English style.
Specifics:
Abstract
Lines 27-29 please re write and improve English structure,
Introduction
Line 49, delete, why will you have subtitles in the introduction?
Lines 42-48 please update references; there is a continuous debate on the pros and cons from consuming milk fatty acids
Line 72 should say Methods or Methodology
Lines 73-79 provide range of dates or months you used for your search, provide data bases used i.e., scopus, web of knowledge, google scholar?
What do you mean by qualitative and quantitative articles?

Tables 1 and 2. What is milk fatty acid (%), is it milk fat content? Please change accordingly
Check the use of decimals in your tables, also if control diets do not have fish oil or algae then use a “–“ which means the same and use it in all places where you do not have data. For decimals use 3 digits, for example 345, 3.45, 0.34

Reviewer 2 ·

Basic reporting

The review provides great insights about DHA metabolism and synthesis in animal organism. In addition, provides information about the use of DHA-rich products in dairy cows nutrition and its excretion in milk, and other tools to increase DHA amount in animal products. This subject is important nowadays and has been greatly studied, since human health can be improved by such findings. Therefore, I support the further publication of this review, after revision is carefully done. Hence, I recommend major revision.
Language is clear, with minimal errors that can be easily corrected. The introduction part is well-written, concise and clear.
Review is supported by adequate references, although some of them should be revised.
Some parts must be improved, as I stated specifically in the pdf attached.
Information provided in the tables should be revised and clarified by explaining how calculations were done. Please, revise the concepts of "milk fat content" and "milk fatty acid content".

Experimental design

Review brings important information, although some should be revised as stated.
Sources are adequately cited, with exception of some that I recommended to be revised.
Review is organized logically in adequate subsections and paragraphs.

Validity of the findings

One of the main problems is the information shown in the tables 1 and 2, where DHA content in milk is calculated based on the papers cited. It appears to be a confusion with "milk fat content" and "milk fatty acid content". Some inferences are made based on this results, so they must be revised before publication. I believe the calculus can not be done on this way to be published. However, if the author provides a feasible explanation and show in details below the table how the calculation was done, this data may be approved.

Additional comments

Review brings relevant information and after major revision. Please, read comments in the pdf attached.

Annotated reviews are not available for download in order to protect the identity of reviewers who chose to remain anonymous.

---

## Round 0.2 · accepted · Accept

Authors revised it considering all comments and suggestions from reviewers, and the revised version reads much better.